# (MINT: Causally Tracing Information Fusion in Multimodal Large Language Models

## Abstract

Multimodal Large Language Models (MLLMs) have demonstrated impressive performance on tasks that involve understanding and integrating information across different modalities, particularly vision and language. Despite their effectiveness, the internal representations of these Vision Language Models (VLMs) remain poorly understood, making it difficult to interpret their predictions or identify the causes of common errors. A crucial step toward improved interpretability is understanding how visual and textual signals fuse within the language decoder of these models. This integration process is particularly important since failures to properly combine modalities frequently lead to errors such as object hallucinations and incorrect spatial descriptions. In this paper, we systematically investigate the internal mechanisms of multimodal fusion in three representative VLMs: LLaVA-1.5-7B, DeepSeek-VL2-Tiny, and Qwen2-VL-7B. We propose MINT (Multimodal INtervention Tracing), a method that builds on the principle of hidden state patching (Ghandeharioun et al., 2024) to create a causal map of multimodal processing by systematically intervening at each layer of the language decoder. From these maps, we identify a critical region we term the 'fusion band'—the decisive window of layers where visual and linguistic signals are actively fused to guide the model's output. Our analysis reveals that the location and width of this band are not uniform across models; they highlight fundamental differences in their fusion mechanisms that directly correlate with a model's ability to resolve contradictions, ground language, and perform complex spatial reasoning. This causal mapping offers a diagnostic framework to explain common VLM failures. Finally, we validate the practical utility of MINT by demonstrating a surgical LoRA fine-tuning strategy that targets the identified fusion band, correcting hallucination failures with near-perfect efficiency using less than half the parameters of baseline approaches.

## 1 Introduction

Multimodal Large Language Models (MLLMs) mark a significant advancement toward AI systems capable of processing and reasoning across diverse input modalities, including text, images, video, and audio (Lu et al., 2019; Tan & Bansal, 2019; Radford et al., 2021; Li et al., 2022; Gu et al., 2023; Wang et al., 2024; Liu et al., 2025). By integrating multiple modalities, MLLMs offer the potential for more human-like perception and reasoning. However, their growing complexity poses challenges in understanding how information from different modalities is infused within the model.

In this work, we focus on VLMs (Lu et al., 2019; Tan & Bansal, 2019; Radford et al., 2021), a core subclass of MLLMs that integrate images and text. VLMs have achieved remarkable performance on tasks such as image captioning (Vinyals et al., 2015; Xu et al., 2016; Karpathy & Fei-Fei, 2015), visual question answering (Agrawal et al., 2016), and compositional reasoning (Johnson et al., 2016; Hudson & Manning, 2019). Yet the internal mechanisms by which visual and textual information are blended remain largely opaque, hindering our ability to interpret model decision-making and identify potential failure modes. Understanding multimodal information fusion within the hidden representations of VLMs is crucial, as improper or misaligned fusion can lead to hallucinations (e.g., nonexistent objects in a scene appear in the model's outputs) (Chen et al., 2024; Geigle et al., 2024; Alhamoud et al., 2025). This poses significant risks in safety-critical domains like medicine (Brin

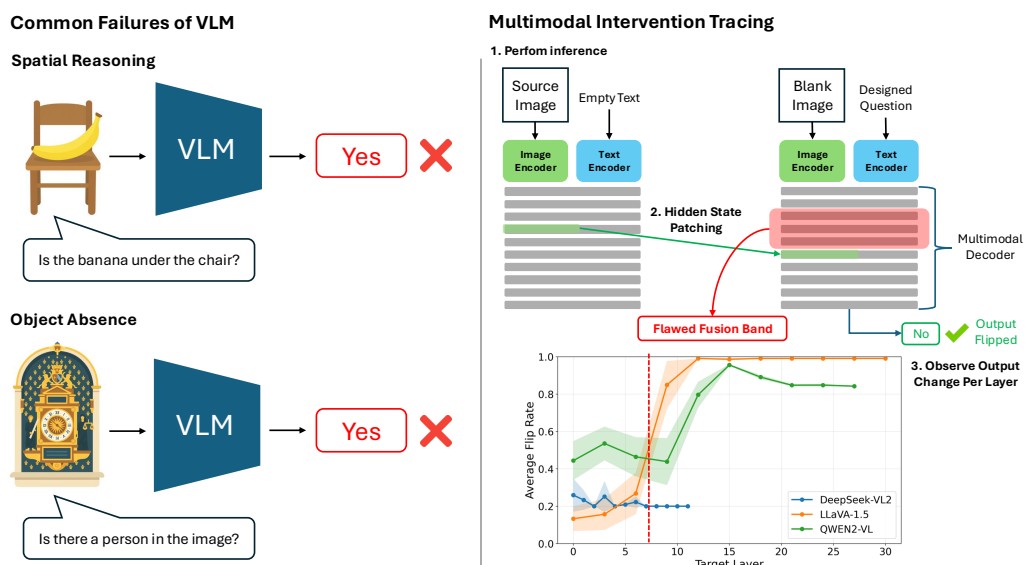

Figure 1: **Our diagnostic framework for VLM failures. (Left)** VLMs often fail on tasks like spatial reasoning and object absence detection. **(Right)** We diagnose these errors by patching a clear visual hidden state from a *source run* into a *target run*. This intervention bypasses a flawed fusion band to causally correct the output. Plotting the correction rate (*Average Flip Rate*) per layer reveals distinct failure modes: some models show a 'late activation' of reasoning abilities (LLaVA-1.5, Qwen2-VL), while others exhibit a more fundamental representational failure (DeepSeek-VL2).

et al., 2024), autonomous systems (Guo et al., 2024; Wu et al., 2025), security surveillance (Yang et al., 2025; De Silva et al., 2025), and legal evidence analysis (Dempsey, 2023; Grossman et al., 2023; Lee et al., 2024).

Interpreting VLMs is inherently challenging. Unlike unimodal models where attention can be traced token-to-token (Bahdanau et al., 2016), VLMs fuse heterogeneous signals like text tokens and visual patches within opaque, high-dimensional hidden states. This complex fusion makes it difficult to disentangle each modality's contribution, creating a significant barrier to attribution and diagnosis (Hendricks et al., 2021; Abnar & Zuidema, 2020). Therefore, our central question is: **Where and how do visual and textual signals interact within the decoder of VLMs to influence generation?**

Previous work on interpreting VLMs has primarily focused on probing analyses (Conneau et al., 2018; Hewitt & Manning, 2019), which train lightweight classifiers on frozen representations to reveal what information is encoded at different layers. While effective at revealing representational content, these methods are correlational; they can show that certain information is encoded in a model's hidden states but cannot demonstrate if that information causally influences the model's final output. To move beyond correlation and establish a direct causal link between internal states and behavior, recent work has applied causal interventions to VLMs; however, these explorations remain limited in scope. Initial studies on models like BLIP centered on the **vision encoder** (Palit et al., 2023; Golovanevsky et al., 2025), which is insufficient for modern architectures where critical fusion occurs in the **language decoder**. More recent work on models like LLaVA (Liu et al., 2024) has employed narrow interventions such as **editing attention mechanisms** (Wang et al., 2025) or **token ablation** (Neo et al., 2025). Although powerful, these methods do not reveal how complete, layer-wise hidden states are integrated during generation. Consequently, a holistic, causal map of how multimodal information is fused *within the language decoder* remains missing.

To fill this gap, we propose a method, MINT (Multimodal INtervention Tracing), that builds upon the principles of hidden state patching, a form of causal intervention systematically explored in frameworks like Patchscopes (Ghandeharioun et al., 2024). MINT is the first to apply hidden state patching technique to causally probe the language decoders of modern VLMs, enabling the construction of a precise map of the fusion layers where visual and textual signals converge. This approach identifies the semantic midpoints of the fusion process, providing not only diagnostic insight into

failures like hallucination and spatial errors but also a **roadmap for targeted model correction** (see Figure 1).

The main contributions of this work are three-fold. (1) We introduce a systematic framework for causally interpreting VLMs by adapting hidden state patching to the multimodal setting, allowing for precise, layer-wise analysis of the information fusion process. (2) We provide the first empirical map of the "fusion band" across distinct VLM architectures, revealing that the location and width of this band are key architectural differentiators. (3) We demonstrate the **actionable utility** of our framework by showing that surgical fine-tuning of the identified fusion band can correct hallucination failures with significantly higher parameter efficiency than broad baseline approaches.

## 2 RELATED WORK

Early interpretability efforts in VLMs have predominantly relied on probing analyses (Conneau et al., 2018; Hewitt & Manning, 2019). These methods train lightweight classifiers on a model's frozen internal representations to reveal what information is encoded at various layers. Probing has been applied to investigate a range of properties in VLMs, including positional information (Rösch & Libovický, 2022), conceptual understanding (Schiappa et al., 2024), and representational power (Cekinel et al., 2024). While effective at mapping the representational content of hidden states, these correlational techniques fall short of demonstrating whether the encoded information actively drives the model's generation process. This limitation leaves a critical gap in establishing causal relationships between internal states and a model's observable behavior, motivating the need for interventional approaches.

To address the limitations of correlational methods, recent work has turned to causal intervention techniques like activation patching. The foundational PatchScope framework formalized hidden state patching for transformers, where a state from a "source" run is injected into a "target" run to causally link internal representations to outputs (Ghandeharioun et al., 2024); this technique requires careful design to avoid common pitfalls (Heimersheim & Nanda, 2024). Building on this foundation, researchers have applied these techniques to reveal the internal workings of VLMs. Several studies adapted causal tracing to probe earlier architectures like BLIP, successfully tracing information through the vision encoder (Palit et al., 2023; Golovanevsky et al., 2025). More recent work has introduced finer-grained interventions on modern architectures, such as editing attention heads (Wang et al., 2025) or using token ablation to pinpoint where object information is processed (Neo et al., 2025).

While these approaches provide localized insights into components like encoders or individual attention heads, a systematic, layer-wise map of the end-to-end fusion process within the language decoder remains missing. To fill this gap, our work introduces a causal, layer-wise analysis of the entire decoder, moving beyond component-specific edits to create a holistic diagnostic map of how cross-modal signals converge. By connecting these fusion points to common failure modes, our framework provides a powerful new tool for interpreting and diagnosing VLM behavior.

## 3 METHOD

Our investigation is guided by three specific research questions:

- **Fusion Localization:** Is there a consistent band of decoder layers where visual and textual signals are fused?
- **Causal Role:** How does successful patching into these layers alter outputs, e.g., by overriding linguistic priors with visual evidence?
- **Error Diagnosis:** Can the localization of fusion points explain recurring VLM failures such as hallucinations, spatial confusion, and negation errors?

To answer these questions, Our central goal is to causally identify **where** in a VLM's decoder visual and textual information are fused to shape the final output. To achieve this, we adapt the hidden state patching technique (Ghandeharioun et al., 2024), which allows us to perform controlled interventions during the model's inference process.

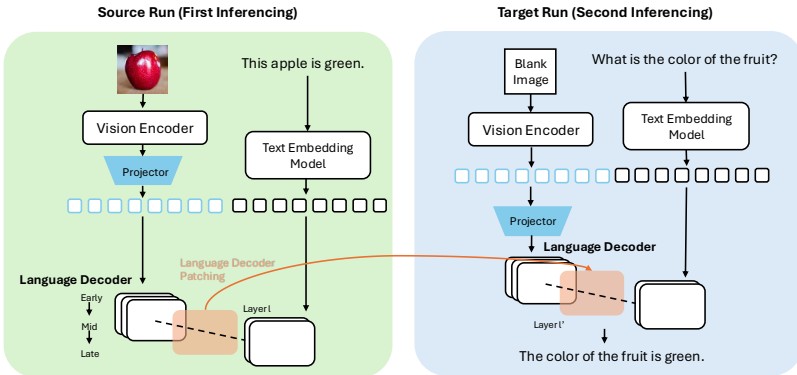

Figure 2: Diagram of our patching framework. Hidden states (e.g., visual, textual, or multimodal) extracted from the source run are injected into the target inference to assess their influence on the model's output.

The framework, illustrated in Figure 2, relies on two model runs: a *source* run from which we extract a hidden state containing specific information (e.g., the visual concept of an "apple"), and a *target* run where the original hidden state is overwritten with the one from the source. By observing how this intervention changes the target's output (e.g., causing the model to output "apple" even when shown a different fruit), we can pinpoint the layers responsible for processing that specific information.

To formalize this, let a Vision-Language Model (VLM) be a function $f_\theta(X_I, X_T)$ that takes an image $X_I \in \mathcal{I}$ and a text sequence $X_T \in \mathcal{T}$. The model is composed of a vision encoder $E_v$, a language encoder $E_l$, and a transformer-based decoder $D_\theta$ of depth $L$. The decoder processes the encoded inputs through a series of layers $\ell \in \{1, \ldots, L\}$. We denote the hidden representation at a specific decoder layer $\ell$ as:

$$h_\ell(E_v(X_I), E_l(X_T)) \in \mathbb{R}^d$$

where $d$ is the hidden dimension.

Given a source input $(X_I^s, X_T^s)$ and a target input $(X_I^t, X_T^t)$, we denote their respective hidden states at layer $\ell$ as $h_\ell^s$ and $h_\ell^t$. The patching operation is detailed in Algorithm 1. The resulting patched output for the target input is:

$$y_{\text{patch},\ell}^t = f_\theta(X_I^t, X_T^t; \text{patched with } h_\ell^s \text{ at layer } \ell).$$

The primary objective of our analysis is to characterize the set of layers $\Lambda$, which we define as the "fusion band":

$$\Lambda = \left\{ \ell \mid y_{\text{patch},\ell}^t \neq f_\theta(X_I^t, X_T^t) \right\},$$

This set contains all layers where substituting the source's hidden state causally alters the target's original output. Identifying $\Lambda$ provides a direct map of where multimodal fusion occurs and enables us to diagnose how failures like hallucination or spatial errors arise from misaligned representations. The output of the framework is the altered text generation, which we analyze using metrics like **Override Accuracy** and **Flip Rate** (detailed in Section 4.2) to quantify the causal effect at each layer.

## 4 EXPERIMENTAL SETUP

This section details the experimental setup for our investigation. We first introduce the three representative VLMs and the benchmark datasets used to probe their internal mechanisms. We then define the set of evaluation metrics employed to quantify the effects of our patching interventions. This setup provides the foundation for the diagnostic analysis presented in Section 5.

## 4.1 MODELS AND DATASETS

We evaluate our framework on representative open-source VLMs that span different architectural choices. LLaVA-1.5-7B (Liu et al., 2024) combines a CLIP ViT-L/14 visual encoder with a Vicuna-7B language model, and has become a widely used baseline for multimodal instruction following. DeepSeek-VL2-Tiny (Wu et al., 2024) adopts a SigLIP vision encoder paired with a lightweight 2B-parameter decoder, providing a compact yet efficient contrast. Qwen2-VL-7B (Bai et al., 2023) represents the Qwen family's latest release, with a strong decoder and high-resolution visual perception. Additionally, to capture the latest architectural evolutions, we extend our analysis to the state-of-the-art Qwen2.5-VL-7B. For all models, we restrict interventions to the decoder, sampling layers at regular intervals to balance interpretive granularity with computational efficiency.

Evaluations are carried out on curated subsets of multimodal benchmarks. To study spatial reasoning, we use the What'sUp dataset (Kamath et al., 2023), which provides controlled counterfactual pairs for geometric relations. For probing visual negation, we adapt NegBench (Alhamoud et al., 2025), a benchmark explicitly designed to test object absence and contradiction in multimodal reasoning. Finally, for general object identification and caption grounding, we sample images from MS COCO (Lin et al., 2015), a large-scale dataset widely used in vision–language research. For each experiment, we construct paired source–target inputs aligned with our probing objectives, ensuring that patching outcomes can be causally interpreted as evidence of where and how fusion occurs within the decoder. Note that for experiments employing a "blank image" baseline, we use a black image tensor (zeros) processed through the vision encoder. This strictly preserves input dimensions and sequence lengths while removing visual semantic information.

## 4.2 EVALUATION METRICS

To quantify the causal effect of our patching interventions, we employ three complementary metrics that together provide a comprehensive view of the model's internal processing. First, we measure *Override Accuracy*, which captures the percentage of cases where a patched hidden state causes the model's output to align with the source inference, thereby overriding the original target prediction. While this metric indicates whether source representations dominate the output, it does not by itself distinguish between correcting failures and reinforcing already correct predictions. To address this, we further track the *Flip Rate*, defined as the fraction of baseline failures in the target that are corrected after patching. This measure directly reflects how effectively injected representations repair systematic weaknesses in the model. Finally, we introduce *Failure Depth*, which identifies the earliest decoder layer at which the median Flip Rate across source layers exceeds a fixed threshold (set to **0.5**). This metric highlights the depth within the decoder where visual evidence begins to consistently override linguistic priors, thereby revealing where causal influence first emerges in the model's hierarchy.

## 5 MAPPING THE PATHWAYS OF MULTIMODAL FUSION

To answer our core research questions regarding fusion localization, causal roles, and error diagnosis, we conducted a series of five targeted patching experiments. Each experiment is designed to isolate a specific aspect of the vision-language fusion process, from foundational mechanisms to the sources of common failures in advanced reasoning. Table A.4 provides a high-level overview of these experiments, outlining the primary goal and methodology of each probe. The following subsections detail the setup and rationale for each experiment in full.

## 5.1 WHERE IS VISUAL INFORMATION INTEGRATED INTO THE GENERATIVE PROCESS?

Recent analyses reveal that visual contributions are concentrated in the shallow–middle layers of large vision–language models, where image tokens are fused into the text stream, while the influence of vision decreases in deeper layers, leaving later representations increasingly shaped by linguistic context (Zhang et al., 2025; 2024; Jiang et al., 2025). To causally test this hypothesis and pinpoint the fusion band, we designed a visual override experiment. The core question is: *At which decoder layer has the visual information from the target input been so deeply integrated that a contradictory patched state can no longer alter the output?*

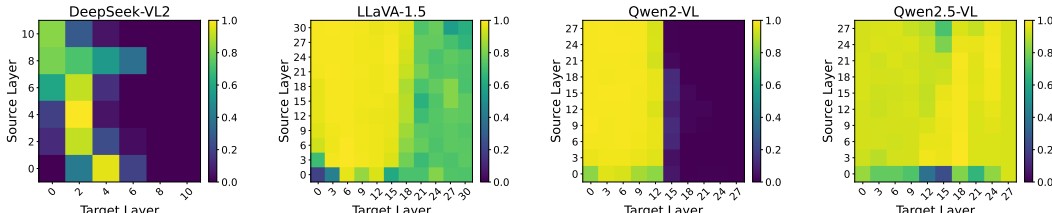

Figure 3: Full source-target heatmaps from the Visual Override experiment. Yellow indicates high Override Accuracy (successful causal intervention). While DeepSeek-VL2, LLaVA-1.5, and Qwen2-VL show distinct, localized "fusion bands" where visual influence drops off, the newer **Qwen2.5-VL** (right) exhibits *Sustained Visual Integration*, remaining causally responsive to visual patches throughout nearly the entire decoder.

We use a forced-choice task where the **source input** $(X_I^s, X_T^s)$ pairs an image of a generic object ([Object B]) from the MS COCO dataset (Lin et al., 2015) with the classification prompt, "What's inside the image? An apple or a [Object B]?". The **target input** $(X_I^t, X_T^t)$ uses an image of an apple with the same prompt. The intervention at each decoder layer $l$ involves replacing the target's hidden states, which are processing the "apple" image, with the corresponding states from the source run containing information about "[Object B]".

A low Override Accuracy $OA(l)$ at a given layer indicates that the patched visual signal failed to steer the model's output. We interpret this failure as evidence that the visual information from the target input (the apple) has already been decisively fused into the model's representation. Once this fusion occurs, the internal states have solidified, making them robust to the contradictory "[Object B]" information from the source patch. Therefore, identifying the layers where Override Accuracy drops off allows us to map the critical 'fusion band' where the model commits to its visual evidence.

This experiment revealed that visual influence is typically concentrated in a distinct **"fusion band"**, whose characteristics vary significantly across architectures (Figure 3). As aggregated in Figure 4, **Qwen2-VL** performs an **early and broad** fusion (solidifying by layer 12), while **LLaVA-1.5** adopts a **late and distributed** strategy (starting around layer 18), and **DeepSeek-VL2** utilizes a **highly localized** mechanism peaking at layer 4. Strikingly, extending MINT to the newer **Qwen2.5-VL** reveals a distinct evolution towards *Sustained Visual Integration*: unlike its predecessors, it maintains high override accuracy across nearly the entire decoder depth (up to layer 27), suggesting that SOTA architectures are shifting towards maintaining high visual plasticity throughout the generation process to avoid early commitment bottlenecks.

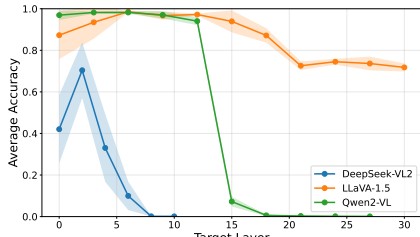

Figure 4: Average override accuracy per target layer, revealing distinct fusion strategies across the three models.

## 5.2 WHEN DO TEXT EMBEDDINGS BECOME VISUALLY GROUNDED?

Having established where direct visual signals influence generation, we now investigate the indirect pathway: how visual information is absorbed and encoded within the text representations themselves. This "text-only patching" experiment is designed to identify the decoder layer at which textual hidden states become sufficiently "visually grounded" to carry meaningful information about the image, even in its absence.

The **source input** $(X_I^s, T)$ consists of a real image, $X_I^s$, containing a specific object, paired with the associated prompt, $T$: "Is there a <category> in the image? Only answer with yes or no.", where '<category>' is the name of the object in $X_I^s$. The **target input** $(X_I^t, T)$ then uses a blank image placeholder, $X_I^t$, while keeping the prompt identical to the source run.

The intervention here focuses on the language stream. At a given decoder layer $l$, we extract the visually-conditioned **textual hidden states** from the source run. These representations are then injected into the corresponding token positions of the decoder during the target inference.

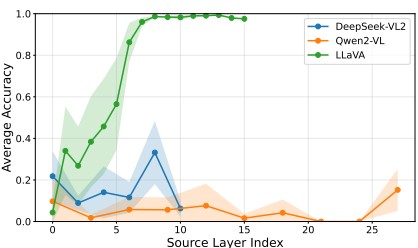

Figure 5: Text Grounding experiment: Average injection accuracy per target layer, revealing a weak, early-layer signal.

The rationale for this setup is to isolate the information encoded in the text embeddings. Since the target run has no visual input, the model's ability to answer "yes" can only stem from the visual semantics preserved within the patched textual states. We measure this with **Injection Accuracy**, defined as the proportion of outputs that correctly answer "yes". A high accuracy at layer $l$ indicates that the text representations have successfully absorbed and retained the visual context from the source image, effectively becoming visually grounded.

The results (Figure 5) reveal a striking architectural divergence. **LLaVA-1.5** exhibits a strong "information absorption" phenomenon: its textual hidden states become visually grounded rapidly, rising from Source Layer 2 and reaching near-perfect effectiveness ($> 95\%$) by Layer 8. This suggests LLaVA's decoder actively "bakes" visual context into text tokens early in the generation process.

In sharp contrast, **Qwen2-VL** and **DeepSeek-VL2** show minimal text grounding effectiveness ($< 20\%$) across all layers. This indicates that these architectures likely maintain a cleaner separation between modalities, relying on continuous attention to visual tokens (via cross-attention or specialized streams) rather than compressing visual semantics into the linguistic state.

### 5.3 How Are Contradictions Between Modalities Resolved?

To isolate how the model prioritizes competing signals, we designed an experiment that places visual evidence and textual description in direct contradiction. At each decoder layer $l$, we patch the **entire hidden state** ($h_{s,l}$) from a **source run**—which pairs an image of an object [Object C] with a contradictory prompt (e.g., "The main object is not [Object C]")—into a neutral **target run** with a blank image. This intervention tests which signal within the *fused source state* dominates subsequent processing. We measure this with **Override Accuracy**, where a "yes" answer indicates that the visual evidence for [Object C] successfully overrode the explicit textual negation.

The results, which show the average override accuracy per source layer in Figure 6, reveal that models resolve cross-modal contradictions using highly distinct strategies. Rather than a single, shared pattern, each model

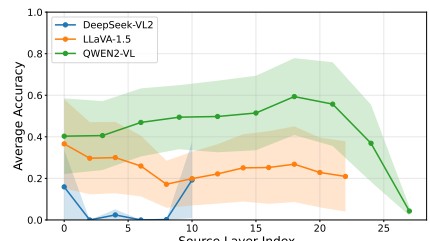

Figure 6: Average override accuracy per source layer for the contradiction experiment, revealing divergent resolution strategies across models.

displays a unique profile for how and when it arbitrates conflicting signals: **Qwen2-VL** shows a process of steadily strengthening its visual conviction, climbing to a distinct peak late in the decoder around **source layer 18**. **LLaVA-1.5** establishes a more consistent, low-to-moderate ability to resolve the conflict that is sustained throughout the decoder. In sharp contrast, **DeepSeek-VL2** exhibits a highly specialized, switch-like behavior; after showing little ability to resolve the conflict in initial layers, a sudden spike in accuracy at **source layer 9** points to a specific mechanism responsible for decisively resolving the contradiction.

### 5.4 Diagnosing Failures in Advanced Reasoning

Beyond mapping foundational mechanisms, our framework serves as a powerful diagnostic tool for investigating well-documented VLM failures. We now apply MINT to diagnose the architectural origins of two critical weaknesses: flawed geometric reasoning and the failure to understand object

absence (negation). For both probes, we measure the **Flip Rate**, the fraction of baseline failures that are causally corrected by patching in a clean visual representation. We generate this representation using a source run with an empty text prompt to ensure the visual signal remains unconditioned by the biasing question. A high flip rate at a specific layer provides strong evidence that the reasoning failure originates in the processing of the layers preceding it.

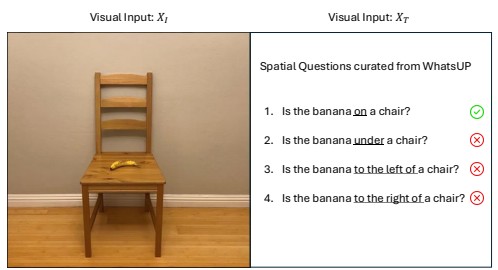

Figure 7: An example from the What'sUp dataset (Kamath et al., 2023), which uses controlled, counterfactual questions to diagnose spatial reasoning.

First, to investigate spatial reasoning, we used image-question pairs from the *What'sUp* dataset(Kamath et al., 2023), illustrated in Figure 7. The results, shown in Figure 8a, reveal two distinct modes of failure. **LLaVA-1.5** and **Qwen2-VL** exhibit a **"late activation"** of their reasoning circuits; their ability to correctly process spatial relations only comes online in the mid-decoder, with the Flip Rate climbing sharply after an initial period of failure. In sharp contrast, **DeepSeek-VL2** shows a more **fundamental representational failure**, with its low Flip Rate across all layers indicating that patching in correct visual information is insufficient to fix its errors.

Next, we probed the model's grasp on negation using the *NegBench* dataset (Alhamoud et al., 2025), which tests the ability to identify object absence. **LLaVA-1.5** and **DeepSeek-VL2** proved robust, showing perfect baseline accuracy on this task. **Qwen2-VL**, however, frequently hallucinated absent objects at baseline.

Our intervention successfully corrected these failures, as illustrated in Figure 8b. The Flip Rate (rate of correction) climbs steeply after layer 9 and reaches a sharp peak at layer 15. This peak at **layer 15** is a critical diagnostic marker, indicating that the hallucination originates from faulty visual-semantic fusion in the shallow layers (before layer 15). Strikingly, this is the same layer where Qwen2-VL's geometric reasoning abilities "activate". The convergence on layer 15 as an effective intervention point for two distinct reasoning tasks strongly suggests a shared, advanced visual-semantic processing hub. This demonstrates our framework's ability to move beyond simply identifying errors to pinpointing their specific architectural origins.

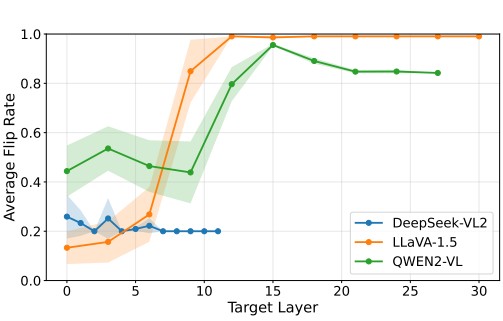

(a) Diagnostic results for geometric reasoning, revealing a "late activation" of spatial abilities (LLaVA, Qwen2) versus a fundamental failure (DeepSeek-VL2).

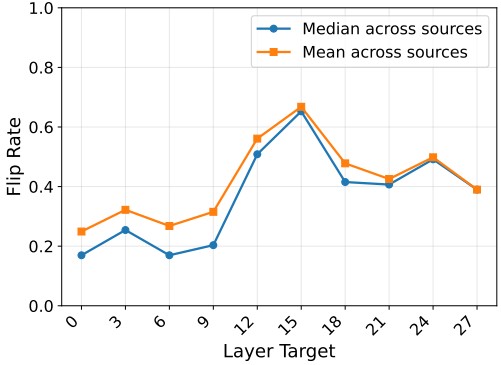

(b) Aggregated Flip Rate for the Object Absence experiment on Qwen2-VL, showing a peak correction rate at layer 15. Full heatmap is in Appendix B.4.

Figure 8: Combined diagnostic results for spatial reasoning and object absence detection.

### 5.5 FROM DIAGNOSIS TO REPAIR: MINT-GUIDED SURGICAL FINE-TUNING

A critical question arising from our diagnostic findings is whether the identified fusion hubs are merely observational correlates or actionable control points. If the "Fusion Band" localized by

MINT is indeed the causal bottleneck for reasoning, then targeting this specific region for intervention should yield more effective model repairs than broader, non-specific approaches.

**Experimental Setup.** To validate this, we conducted a surgical repair experiment on Qwen2-VL, specifically targeting the object absence hallucination failures identified in Section 5.4. We constructed a targeted dataset derived from the probing benchmark used in Section 5.4. First, we identified instances where the base model failed (i.e., hallucinating "Yes" for absent objects). To prevent the model from overfitting to the negative class, we balanced these failures with positive samples (where the object is present). This balanced dataset was then split into training and held-out testing sets. We employed Low-Rank Adaptation (LoRA) with a rank of $r = 8$, targeting the query and value projections (`q_proj`, `v_proj`) within the attention blocks. We compared two distinct strategies:

- **MINT-Guided Repair:** Targeting the LoRA modules exclusively to the critical "Fusion Band" identified by our framework (Layers 14–16).
- **Shallow Baseline:** Targeting a broader range of shallow layers (Layers 1–7), serving as a control to test if performance gains are simply due to adding trainable parameters to early processing stages.

**Results.** We evaluated the models on two metrics: *Correction Rate*, defined as the percentage of previously failed instances in the held-out test set that were successfully repaired; and *General Accuracy*, measured on the full evaluation benchmark from Section 5.4 (5,000 samples) to ensure model robustness. As shown in Table 1, the MINT-Guided strategy achieved a **perfect correction rate of 100.00%** on the test set. Crucially, this performance was achieved using **less than half the trainable parameters** ($1\times$ vs. $2.33\times$) of the Shallow Baseline, which only reached a 77.63% correction rate. Furthermore, the MINT-guided model maintained a general accuracy of 99.98%, confirming that the surgical intervention did not induce catastrophic forgetting.

These results empirically demonstrate that MINT correctly identifies the most efficient architectural bottleneck. By enabling researchers to focus interventions on the causally decisive layers, MINT transforms interpretability from a passive diagnostic exercise into an active tool for surgical model improvement.

Table 1: Comparison of LoRA fine-tuning strategies for correcting object hallucination in Qwen2-VL. The MINT-Guided strategy (targeting the identified Fusion Band) achieves perfect correction with fewer parameters than the Shallow Baseline.

| LoRA Strategy | Layers | Params | Correction Rate | General Acc. | Gain |
|---|---|---|---|---|---|
| Original Model | N/A | N/A | 0.00% | 94.04% | - |
| Baseline (Shallow) | 1–7 | $2.33\times$ | 77.63% | 94.36% | +0.32% |
| **MINT-Guided** | **14–16** | **$1\times$** | **100.00%** | **99.98%** | **+5.94%** |

## 6 Discussion and Conclusion

### 6.1 Synthesis of Experimental Findings

Our investigation reveals that information fusion in VLMs is not a monolithic process but a highly structured mechanism governed by architectural choices. We hypothesize that the location of the identified "fusion band" is structurally determined by the complexity of the vision-language connector. Models with simple connectors, such as LLaVA-1.5's MLP, likely pass raw visual features that require the decoder's early layers to perform preliminary processing and alignment. This architectural constraint delays the effective integration of modalities, resulting in the observed "Late Fusion" pattern. Conversely, sophisticated adapters like Qwen2-VL's C-Abstractor perform extensive pre-processing and feature compression. We posit that these adapters deliver highly aligned features that the decoder can integrate immediately, leading to the "Early Fusion" signature we identified.

Our diagnostics further highlight a critical trade-off between fusion efficiency and reasoning robustness. DeepSeek-VL2 exhibits a highly localized and early fusion band (peaking at Layer 4). While

efficient, this "early commitment" appears to come at a cost: the model's representations harden too quickly, lacking the redundancy needed for complex spatial reasoning. In contrast, Qwen2-VL's broader fusion window allows for sustained visual plasticity, which appears crucial for handling advanced reasoning tasks involving negation and geometry.

## 6.2 Implications for VLM Design

These findings carry significant implications for the design of future VLMs. The evolution from Qwen2-VL to the newer **Qwen2.5-VL** (as analyzed in our supplementary experiments) reveals a shift towards "Sustained Visual Integration," where visual interventions remain effective throughout nearly the entire decoder depth. This suggests that state-of-the-art architectures are moving away from localized fusion towards maintaining high cross-modal plasticity across all layers.

For safety and trustworthiness, the discovery of "snap judgment" mechanisms (like DeepSeek's early commitment) is critical. If a model commits to a visually-driven interpretation in its earliest layers, it may stubbornly resist subsequent textual corrections. Understanding this mechanism is the first step toward designing interventions—such as the surgical LoRA fine-tuning demonstrated in Section 5.5—to mitigate such biases and repair specific reasoning failures.

## 6.3 Limitations and Future Directions

Our study is limited to three open-source models on controlled, discriminative tasks, and our layer-wise patching localizes *where* fusion occurs without identifying the responsible sub-layer components (e.g., attention heads). Future work should therefore apply our framework to a wider VLM taxonomy and use component-level patching for more granular analysis. Most excitingly, our findings enable targeted model repair; for example, one could remediate Qwen2-VL's identified spatial and negation fragility by directly fine-tuning the critical mid-decoder hub we located around layer 15 (Sections 5.4). Such an effort would transition interpretability from a passive act of observation to an active tool for building more robust and reliable multimodal AI.

## 6.4 Conclusion

In this work, we introduced MINT, a systematic framework for causally mapping the information fusion pathways within the decoders of Vision-Language Models. Our application of this framework to three distinct architectures has demonstrated that multimodal integration is not a uniform process but is instead characterized by unique architectural signatures, such as the location and width of a critical *fusion band*. By moving beyond correlational analysis to causal intervention, we have provided a diagnostic lens that can pinpoint the origins of common failures in complex tasks like spatial reasoning and negation. The identification of specific reasoning hubs transforms interpretability from a passive act of observation into an active tool for targeted model improvement. These findings underscore the potential of causal diagnostics to bridge interpretability and robustness, motivating their integration into the design and evaluation of next-generation VLMs.

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

# A APPENDIX

## A.1 USE OF LARGE LANGUAGE MODELS

In the preparation of this manuscript, we utilized Large Language Models (LLMs) as tools to assist with specific aspects of the writing and research process. Our use was guided by the principles of enhancing clarity and productivity while ensuring full authorial responsibility for the final content. The specific applications are detailed below.

### WRITING AND POLISHING

We employed **Google's Gemini 2.5 Pro and OpenAI's GPT-5** to improve the readability and grammatical accuracy of the manuscript. The primary uses included:

- **Paraphrasing and Rephrasing:** To refine sentence structures for better clarity and flow.
- **Grammar and Style Correction:** To identify and correct grammatical errors, typos, and stylistic inconsistencies.
- **Improving Conciseness:** To help condense complex sentences and paragraphs without losing critical information.

LITERATURE RETRIEVAL AND DISCOVERY

LLMs were used to support the literature review process. This involved:

- **Summarizing Articles:** Generating concise summaries of existing research papers to quickly assess their relevance to our work.

- **Brainstorming and Idea Generation:** Exploring related concepts and identifying potential connections between different areas of research to help frame our contributions.

**Authorial Responsibility:** The human authors directed all stages of this research, from conceptualization to the final analysis and conclusions. All text, figures, and results were critically reviewed, edited, and validated by the authors. We take full responsibility for the intellectual content of this paper, including its accuracy, integrity, and any potential errors. The LLMs served as assistive tools and were not given authorship credit.

## A.2 ADDITIONAL ALGORITHMS

In this section, we provide the detailed pseudocode for our patching procedure. Algorithm 1 specifies the single-layer hidden state patching used in our framework. This complements the description in Section 3 by offering a line-by-line view of the intervention.

---

**Algorithm 1** Single-Layer Hidden State Patching for VLM Decoders

---

**Require:** Source input $(I^s, T^s)$, target input $(I^t, T^t)$, a specific decoder layer $\ell$ to patch.
**Ensure:** Patched output $y_{\text{patch}}^t$ from the target inference.
1: **// Source Inference**
2: Run the VLM on $(I^s, T^s)$ and cache the decoder hidden state $h_\ell^s$ at layer $\ell$.
3: **// Target Inference with Intervention**
4: Initialize a forward pass on the target input $(I^t, T^t)$.
5: During the forward pass, when computation reaches layer $\ell$, overwrite the target hidden state:
$$h_\ell^t \leftarrow h_\ell^s$$
6: Continue the forward pass from layer $\ell + 1$ to generate the final output $y_{\text{patch}}^t$.

---

## A.3 MODEL AND DATASET DETAILS

Table 2 summarizes the three VLMs used in our study. We also provide an overview of the probing benchmarks in Table 3. These details extend the descriptions in Section 4.1, giving readers a concise reference.

Table 2: Summary of Vision–Language Models Used in This Study.

| Model | Vision Encoder | Language Decoder | #Params |
|---|---|---|---|
| LLaVA-1.5-7B | CLIP ViT-L/14 | Vicuna-7B | 7B |
| DeepSeek-VL2-Tiny | SigLIP (Tiny) | DeepSeek MoE (1B active) | 3B |
| Qwen2-VL-7B | Qwen2 Visual Adapter | Qwen2-7B | 7B |

Table 3: Benchmarks and Probing Tasks Used in This Study.

| Benchmark | Probing Task | Main Property / Split Size |
|---|---|---|
| What'sUp | Spatial Reasoning | 3,000 pairs, controlled counterfactuals |
| NegBench | Visual Negation | 2,500 pairs, object absence/contradiction |
| MS COCO | Object ID & Captioning | 5,000 samples, general vision–language |

A.4 EXPERIMENTAL SUMMARIES

For completeness, we provide a condensed overview of our diagnostic experiments in Table A.4, which aligns with Section 5.

Table 4: Overview of diagnostic experiments.

| Experimental Goal | Probe / Section | Core Question Addressed |
|---|---|---|
| Understanding Foundational Mechanisms | Visual Override (§5.1) Text Grounding (§5.2) Contradiction (§5.3) | Where does the visual signal dominate? When does text absorb visual meaning? How are cross-modal conflicts resolved? |
| Diagnosing Reasoning Failures | Geometric Reasoning (§5.4) Object Absence (§5.5) | Where does spatial understanding fail? How fragile is the concept of negation? |

## B    SUPPLEMENTAL EXPERIMENTAL RESULTS

This section contains the primary result figures for the experiments discussed in Sections 5.2 through 5.5 of the main paper. These are provided here to supplement the main text.

### B.1    EXPERIMENT 2: TEXT GROUNDING

This section provides the full results for the Text Grounding experiment, discussed in Section 5.2. This experiment isolates the language stream to determine if and when textual hidden states absorb sufficient visual semantic information to answer questions without direct visual input.

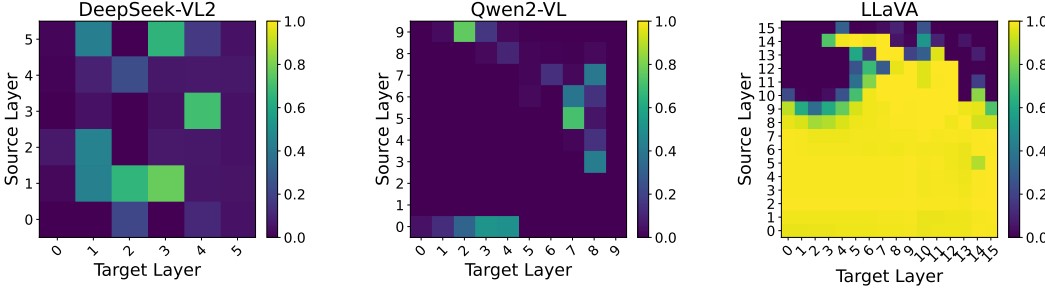

Figure 9: Full source-target heatmaps for the Text Grounding experiment. The y-axis represents the **Source Layer** (where the text state is extracted), and the x-axis represents the Target Layer (where it is injected). **LLaVA-1.5 (Right)** exhibits a broad, high-intensity region, indicating that its textual representations reliably encode visual information starting from Source Layer ∼5. In contrast, **DeepSeek-VL2 (Left)** shows only a faint, localized signal (peaking at Source Layer 3), and **Qwen2-VL (Middle)** shows negligible text grounding, suggesting these models maintain a stricter separation between visual and textual modalities.

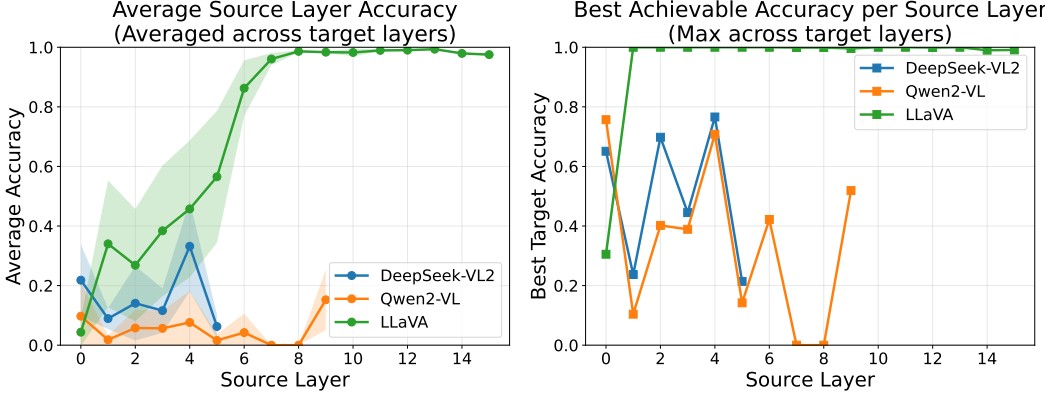

Figure 10: Aggregated performance metrics for Text Grounding. (Left) **Average Source Layer Effectiveness**: This plot reveals the critical architectural divergence. **LLaVA-1.5** demonstrates a strong "absorption" curve, reaching near 100% effectiveness after Source Layer 8. Conversely, **DeepSeek-VL2** and **Qwen2-VL** remain below 20% effectiveness, relying on cross-attention rather than embedding visual semantics into the text stream. (Right) **Best Achievable Accuracy**: The maximum performance per source layer (selecting the optimal target layer), confirming that LLaVA's text embeddings are sufficient to solve the task, while the other models require direct access to visual tokens.

### B.2    EXPERIMENT 3: CONTRADICTION RESOLUTION

This section provides supplemental results for the Contradiction Resolution experiment, detailed in Section 5.3 of the main paper. The goal of this experiment is to understand how models priori-

tize competing signals by patching a fused hidden state containing contradictory visual and textual information into a neutral run.

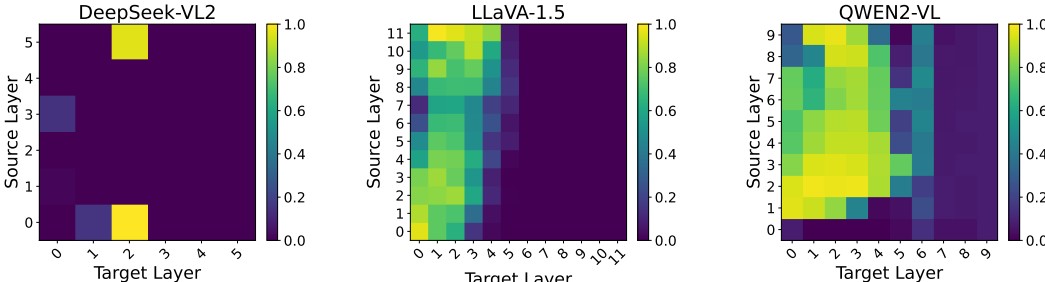

Figure 11: Full source-target heatmaps for the Contradiction Resolution experiment. Each cell shows the Override Accuracy when patching the fused state from a source layer (y-axis) into a target layer (x-axis). The distinct patterns reflect the unique arbitration strategies discussed in the main paper: a sharp, localized resolution for DeepSeek-VL2, and more distributed processing for LLaVA-1.5 and Qwen2-VL.

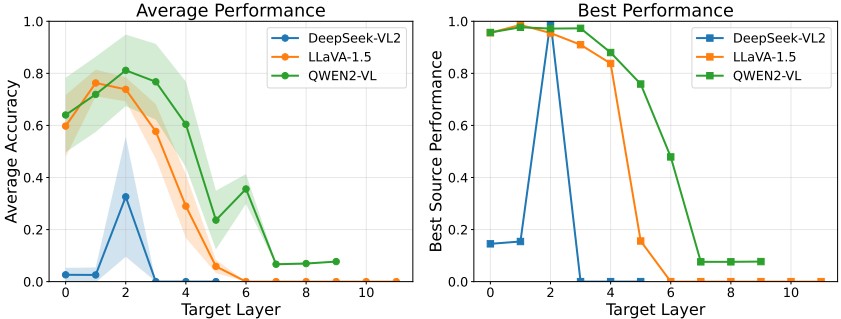

Figure 12: Best Source Performance per target layer for the Contradiction Resolution experiment. This plot shows the maximum possible override effect at each target layer by choosing the optimal source layer. This complements Figure 5 in the main paper by showing where in the decoder the models are most susceptible to having their outputs flipped by a contradictory signal, rather than showing how "decided" each source layer's representation is.

### B.3 EXPERIMENT 4: GEOMETRIC REASONING

This section presents the supplemental results for the Geometric Reasoning experiment, discussed in Section 5.4 of the main paper. This diagnostic experiment investigates the architectural origins of VLM failures on spatial relationship tasks by measuring the "Flip Rate"—the rate at which patching a correct visual representation causally fixes a baseline error.

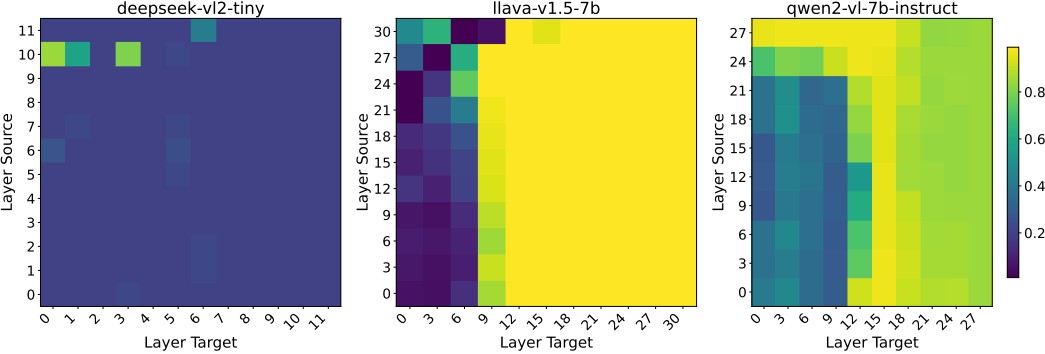

Figure 13: Full source-target heatmaps of the Flip Rate for the Geometric Reasoning experiment. These plots causally diagnose the failure modes described in Section 5.4. LLaVA-1.5 and Qwen2-VL show a distinct vertical band where patching becomes effective mid-decoder, indicating a "late activation" of their reasoning circuits. In contrast, DeepSeek-VL2's heatmap is uniformly low, indicating a more fundamental inability to utilize the patched geometric information.

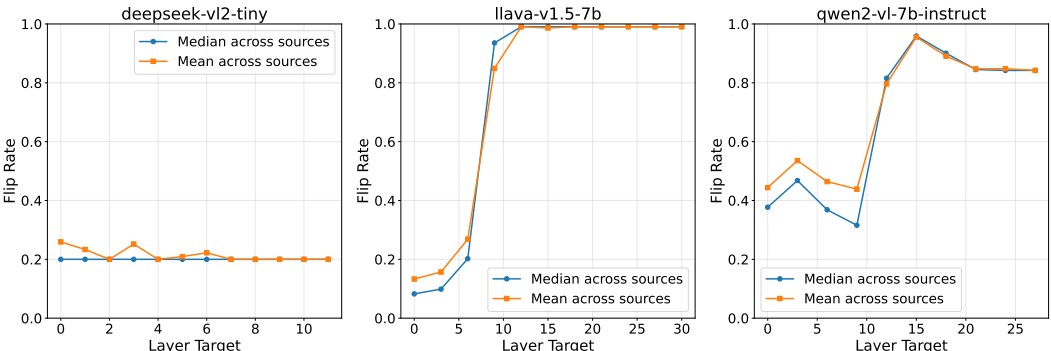

Figure 14: Aggregated Flip Rate (mean and median) per target layer for each model in the Geometric Reasoning experiment. These plots provide a detailed, per-model view of the results summarized in Figure 7 of the main paper. The sharp sigmoidal curves for LLaVA-1.5 and Qwen2-VL clearly show the "activation point" where their spatial reasoning abilities come online. DeepSeek-VL2's flat, low curve confirms its representational failure on this task.

## B.4 EXPERIMENT 5: OBJECT ABSENCE

This section provides supplemental results for the Object Absence experiment, detailed in Section 5.5 of the main paper. This diagnostic experiment investigates the fragility of the models' understanding of negation and non-existence. The results for Qwen2-VL, which exhibited baseline failures (hallucinations), are presented here.

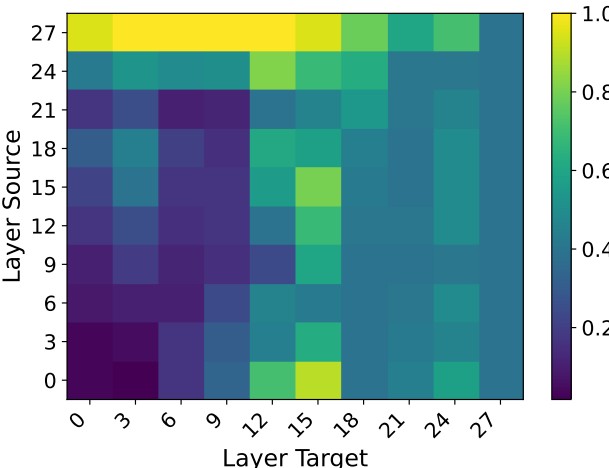

Figure 15: Full source-target heatmap of the Flip Rate for the Object Absence experiment on Qwen2-VL. This plot, originally part of Figure 8 in the main text, shows the rate of correcting a baseline hallucination error. The highest correction rates are concentrated when patching into the mid-decoder layers (approx. 12-18), with a distinct peak at target layer 15. This causally traces the origin of the hallucination to faulty fusion processes occurring in the shallower layers before this point.

# C STATISTICAL SIGNIFICANCE ANALYSIS

This section provides a detailed statistical analysis for each of the five experiments presented in the main paper. For each experiment, we report:

1. Detailed visualizations of the layer-wise intervention effects, including source-target heatmaps with significance markers and aggregated line plots with 95% confidence intervals.

2. A summary bar chart of the overall intervention effect per model.

3. A table with precise metrics, including the mean effect and 95% non-parametric bootstrap confidence intervals (1000 resamples).

## EXPERIMENT 1: VISUAL OVERRIDE

Here we present the statistical analysis for the **Visual Override** experiment. The results confirm that the observed intervention effects are statistically significant for all three models. Figure 16 visualizes the layer-wise and overall effects, while Table 5 provides the precise numerical values.

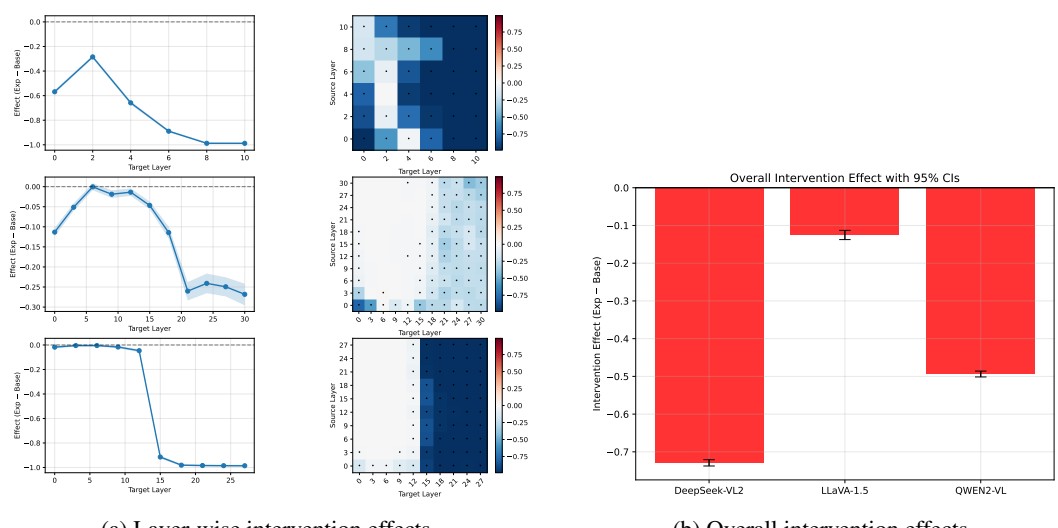

(a) Layer-wise intervention effects.                    (b) Overall intervention effects.

Figure 16: Statistical analysis for the Visual Override experiment. (a) Heatmaps showing the intervention effect across source and target layers for each model. (b) The overall intervention effect for each model, with 95% confidence intervals shown as error bars.

Table 5: Overall metrics and significance for the Visual Override experiment.

| Model | Exp Mean | Exp CI Low | Exp CI Up | Base Mean | Base CI Low | Base CI Up | Effect Mean | Effect CI Low | Effect CI Up | Significant |
|---|---|---|---|---|---|---|---|---|---|---|
| DeepSeek-VL2 | 0.259 | 0.255 | 0.265 | 0.989 | 0.982 | 0.995 | -0.730 | -0.738 | -0.721 | True |
| LLaVA-1.5 | 0.861 | 0.851 | 0.870 | 0.986 | 0.978 | 0.993 | -0.125 | -0.137 | -0.113 | True |
| QWEN2-VL | 0.493 | 0.489 | 0.496 | 0.987 | 0.979 | 0.993 | -0.494 | -0.502 | -0.486 | True |

EXPERIMENT 2: TEXT GROUNDING

Here we present the statistical analysis for the **Text Grounding** experiment. The results confirm that the observed intervention effects are statistically significant for both models. Figure 17 visualizes the layer-wise and overall effects, while Table 6 provides the precise numerical values.

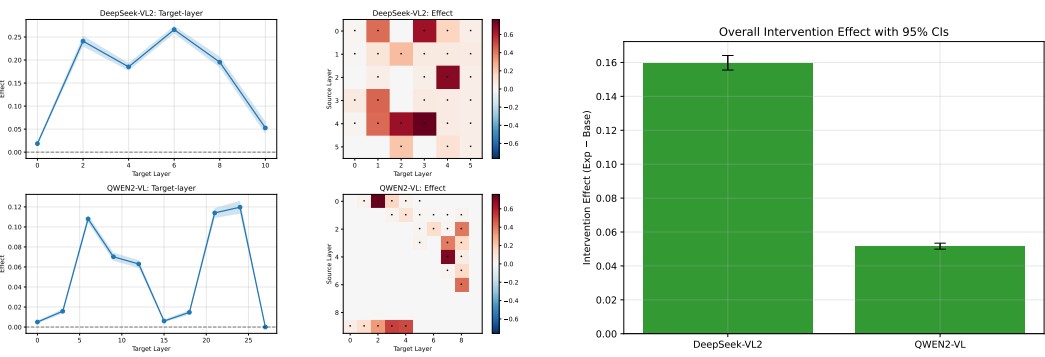

(a) Layer-wise intervention effects.          (b) Overall intervention effects.

Figure 17: Statistical analysis for the Text Grounding experiment. (a) Heatmaps and line plots showing the intervention effect across source and target layers for each model. (b) The overall intervention effect for each model, with 95% confidence intervals shown as error bars.

Table 6: Overall metrics and significance for the Text Grounding experiment.

| Model | Exp Mean | Exp CI Low | Exp CI Up | Base Mean | Base CI Low | Base CI Up | Effect Mean | Effect CI Low | Effect CI Up | Significant |
|---|---|---|---|---|---|---|---|---|---|---|
| DeepSeek-VL2 | 0.160 | 0.156 | 0.164 | 0.000 | 0.000 | 0.000 | 0.160 | 0.156 | 0.164 | True |
| QWEN2-VL | 0.052 | 0.050 | 0.053 | 0.000 | 0.000 | 0.000 | 0.052 | 0.050 | 0.053 | True |

EXPERIMENT 3: CONTRADICTION RESOLUTION

Here we present the statistical analysis for the **Contradiction Resolution** experiment. The results confirm that the observed intervention effects are statistically significant for all three models. Figure 18 visualizes the layer-wise and overall effects, while Table 7 provides the precise numerical values.

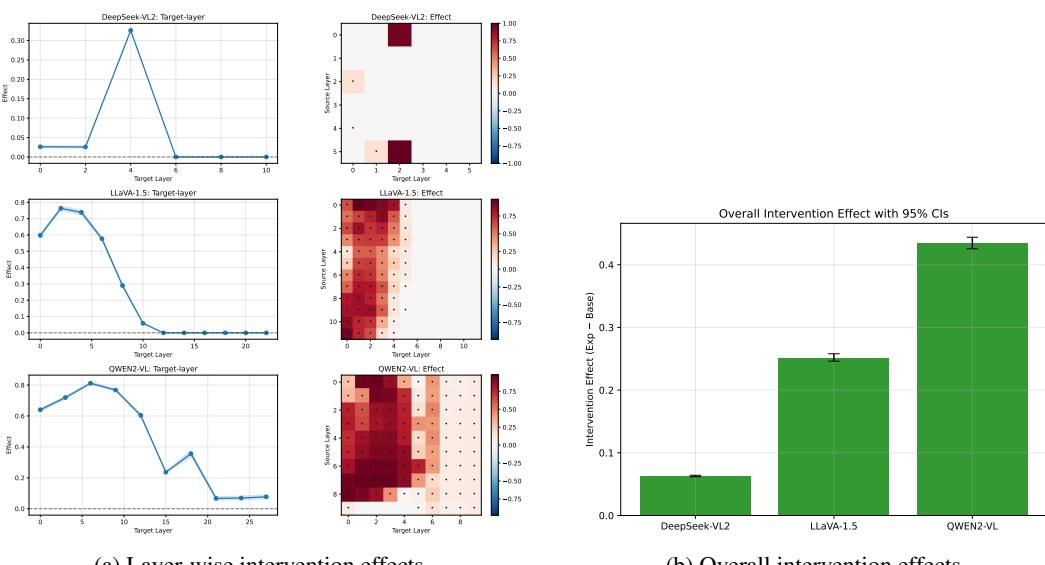

(a) Layer-wise intervention effects.                    (b) Overall intervention effects.

Figure 18: Statistical analysis for the Contradiction Resolution experiment. (a) Heatmaps and line plots showing the intervention effect across source and target layers for each model. (b) The overall intervention effect for each model, with 95% confidence intervals shown as error bars.

Table 7: Overall metrics and significance for the Contradiction Resolution experiment.

| Model | Exp Mean | Exp CI Low | Exp CI Up | Base Mean | Base CI Low | Base CI Up | Effect Mean | Effect CI Low | Effect CI Up | Significant |
|---|---|---|---|---|---|---|---|---|---|---|
| DeepSeek-VL2 | 0.063 | 0.062 | 0.064 | 0.000 | 0.000 | 0.000 | 0.063 | 0.062 | 0.064 | True |
| LLaVA-1.5 | 0.252 | 0.246 | 0.258 | 0.000 | 0.000 | 0.000 | 0.252 | 0.246 | 0.258 | True |
| QWEN2-VL | 0.435 | 0.425 | 0.444 | 0.000 | 0.000 | 0.000 | 0.435 | 0.425 | 0.444 | True |

DIAGNOSING FAILURES IN ADVANCED REASONING

EXPERIMENT 4: GEOMETRIC REASONING

Here we present the statistical analysis for the **Geometric Reasoning** experiment. Figure 19 visualizes the layer-wise and overall effects, while Table 8 provides the precise numerical values.

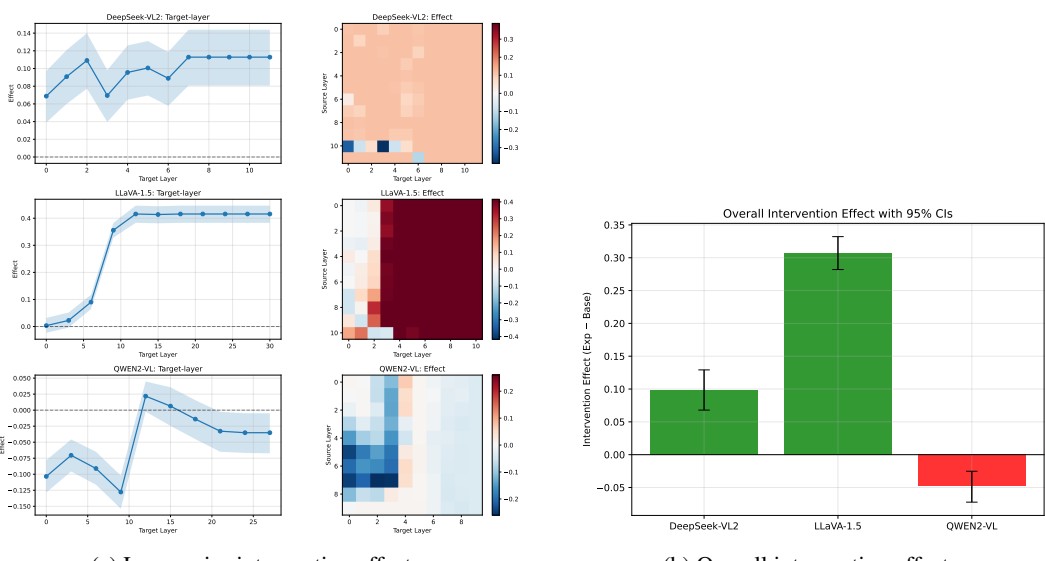

(a) Layer-wise intervention effects.                (b) Overall intervention effects.

Figure 19: Statistical analysis for the Geometric Reasoning experiment. (a) Heatmaps and line plots showing the intervention effect. (b) The overall intervention effect for each model, with 95% confidence intervals.

Table 8: Overall metrics and significance for the Geometric Reasoning experiment.

| Model | Exp Mean | Exp CI Low | Exp CI Up | Base Mean | Base CI Low | Base CI Up | Effect Mean | Effect CI Low | Effect CI Up | Significant |
|---|---|---|---|---|---|---|---|---|---|---|
| DeepSeek-VL2 | 0.736 | 0.717 | 0.755 | 0.637 | 0.614 | 0.660 | 0.099 | 0.068 | 0.129 | True |
| LLaVA-1.5 | 0.642 | 0.631 | 0.653 | 0.334 | 0.311 | 0.357 | 0.307 | 0.282 | 0.332 | True |
| QWEN2-VL | 0.695 | 0.684 | 0.706 | 0.743 | 0.723 | 0.763 | -0.048 | -0.072 | -0.025 | True |

EXPERIMENT 5: OBJECT ABSENCE (NEGATION)

Here we present the statistical analysis for the **Object Absence (Negation)** experiment. Figure 20 visualizes the layer-wise and overall effects, while Table 9 provides the precise numerical values.

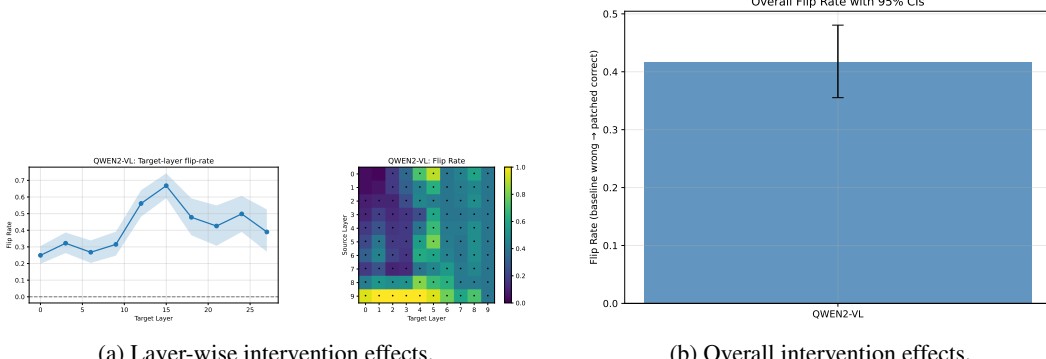

(a) Layer-wise intervention effects.       (b) Overall intervention effects.

Figure 20: Statistical analysis for the Object Absence (Negation) experiment. (a) Heatmaps and line plots showing the intervention effect. (b) The overall intervention effect for each model, with 95% confidence intervals.

Table 9: Overall Flip Rate metrics and significance for the Object Absence (Negation) experiment. This analysis applies only to models that exhibited baseline failures.

| Model | Flip Rate Mean | 95% CI Low | 95% CI Up | Significant |
|---|---|---|---|---|
| DeepSeek-VL2 | Not Applicable (0 baseline failures) | | | |
| LLaVA-1.5 | Not Applicable (0 baseline failures) | | | |
| QWEN2-VL | 0.417 | 0.355 | 0.481 | True |

