# OpenReview forum: "MINT: Causally Tracing Information Fusion in Multimodal Large Language Models"
_ICLR.cc/2026/Conference — ICLR 2026 Conference Withdrawn Submission_

### Official Review · Reviewer_qNLH · 2025-10-30

**Soundness:** 3
**Presentation:** 3
**Contribution:** 3
**Rating:** 6
**Confidence:** 4

**Summary:**

This paper investigates how visual and textual information are fused inside Multimodal Large Language Models (MLLMs) to better understand their internal reasoning and interpret their errors. The authors introduce MINT, a causal analysis method based on hidden-state patching that systematically intervenes in each decoder layer to trace where and how multimodal integration occurs. Applying MINT to LLaVA-1.5-7B, DeepSeek-VL2-Tiny, and Qwen2-VL-7B, the study identifies a “fusion band”—a specific range of layers where visual and textual signals interact most strongly. The paper finds that the position and width of this fusion band vary across models, reflecting distinct fusion mechanisms that correlate with capabilities in grounding, contradiction resolution, and spatial reasoning. Overall, the work provides a causal interpretability framework for diagnosing and comparing VLMs, offering insights for future model design and multimodal understanding.

**Strengths:**

1. The introduced probing method MINT, is systematic and causal method to trace multimodal fusion within VLMs, advancing beyond correlational analyses and offering a concrete tool for understanding internal model mechanisms.

2. By analyzing widely-used MLLMs, the study reveals model-specific fusion patterns, providing generalizable insights into how architectural differences affect grounding and reasoning capabilities.

3. The discovery offers a clear, interpretable indicator of where visual–textual integration occurs, enabling targeted diagnosis of common VLM failures (e.g., hallucination, spatial errors) and guiding future multimodal model design.

**Weaknesses:**

1. Lines 326–328 state that “answering yes can only stem from the visual semantic.” It would be more convincing to empirically verify that the those prompts containing <category name> do not contain textual token biases toward “yes.” In other words, the models should be tested to ensure they do not produce “yes” responses when the visual input is patched with the placeholders.

2. The description of “patching in a clean visual representation” in Section 5.4 is somewhat ambiguous. It would help to clarify whether this refers to the output of the multimodal projector or another specific representation. Clearer definition and justification are needed to support the validity of the intervention results.

3. The conclusion of a “fundamental representational failure” for DeepSeek-VL2 (Line 397) seems overstated. As discussed in Sections 5.2 and 5.3, DeepSeek-VL2 performs visual grounding in the later decoder layers, a behavior distinct from the other two models. Thus, manually patching visual tokens might introduce an excessively strong signal rather than revealing an inherent failure.

4. The analysis would be more convincing if extended to larger models. Observations drawn from smaller models may not generalize to deeper models, potentially limiting the robustness of the conclusions.

**Questions:**

1. Line 318-321 mentioned a blank image placeholder. What placeholder is it? A special token like <vision_pad> or other tokens.


Typos:

1. The  citation formatting is problematic through the whole paper. There should be a pair of brackets between your citations. Please check the latex grammar.

2. In Figure 8 (a), the quotation mark of ``late activation'' is mistaken.

---

> ### Author Response · Authors · 2025-12-01
>
> We sincerely thank the reviewer for their positive assessment, noting MINT as a **"systematic and causal method"** that offers **"concrete tools"** for understanding internal mechanisms. We value their constructive feedback on ensuring the rigor of our baselines and the precision of our definitions.
>
> We address the reviewer's specific concerns and questions below.
>
> **About Language Bias and Control Baselines**
>
> We thank the reviewer for this suggestion. To test for textual bias, we ran the "Blank Baseline" control experiment. The results confirm our findings are not driven by the prompt: with blank images and no patching, LLaVA-1.5 and DeepSeek-VL2 showed a **0% "Yes" response rate**, and Qwen2-VL only **4%**. This minimal baseline noise empirically verifies that the results in our experiments are causally driven by the patched representations, not a systematic textual bias. We have added these findings to the Appendix.
>
> **About Ambiguity of "Clean Visual Representation"**
>
> We confirm "clean" does **not** mean the multimodal projector output. As shown in **Figure 1 (Right Panel)**, "clean" refers to the **decoder hidden state ($h_l^s$)** from a source run. This run uses the same image but a critical control: an **Empty Text Prompt**. This ensures the state encapsulates the processed visual semantics *without* being conditioned by the hallucination-triggering prompt (e.g., "Is there a person?"). This allows us to isolate the true causal effect of the visual information. We will clarify this definition in Section 5.4.
>
> **About Interpreting DeepSeek-VL2's Behavior**
>
> We agree that "failure" is too strong. As shown in our own **Figure 4**, DeepSeek performs fusion extremely early (peaking around Layer 4). The lack of "Flip Rate" in later layers indicates the model has **"hardened" its decision (early commitment)** much earlier than LLaVA or Qwen, making late-stage patching ineffective. This is an architectural distinction (Early vs. Late Fusion) rather than a capability deficit. We will revise the manuscript to characterize this as **"resistance to late-stage causal intervention due to early fusion,"** ensuring a fairer and more accurate architectural comparison.
>
> **Response to Questions and Typos**
>
> **Q1: What is the blank image placeholder?**
> The "**blank image placeholder**" is a black image tensor with all pixel values set to zero. This is processed through the vision encoder like a standard image. We use this method to ensure that the input dimensions, sequence length, and positional embeddings remain strictly consistent between the Source and Target runs, isolating the visual content as the only variable. We will add this definition to the experimental setup.
>
> **Typos:** We have corrected the citation formatting and fixed the quotation mark usage in Figure 8(a) in the updated version.

---

### Official Review · Reviewer_K6zq · 2025-10-30

**Soundness:** 3
**Presentation:** 3
**Contribution:** 2
**Rating:** 4
**Confidence:** 3

**Summary:**

This paper focuses on the interpretability of multimodal information fusion in visual language models (VLMs), proposes MINT method, constructs causal maps through hidden state patching technology, successfully identifies the key fusion region of "fusion band", and verifies the model specificity of the fusion mechanism based on three representative VLMs (LLaVA - 1.5-7 B, DeepSeek-VL2-Tiny, Qwen2-VL-7B). At the same time, it realizes the diagnosis of common faults such as spatial reasoning errors and negative understanding bias.

**Strengths:**

1. MINT first adapts the hidden state patching technology system in single-modal scenes to the language decoder of VLM, breaking through the limitation that traditional probe analysis (such as lightweight classifier probes) can only capture correlations, and directly locates the fusion level of visual-text information through causal intervention, filling the research gap of "multimodal fusion causal map"; the "fusion band" provides a unified index for quantifying the fusion mode of different VLMs, which has strong theoretical significance.
2. The model selection takes into account different architecture types (CLIP + Vicuna, SigLIP + MoE decoder, custom vision adapter + dedicated decoder), and the dataset covers spatial inference, NegBench, and MS COCO, which can comprehensively verify the performance of the fusion mechanism in different tasks. The evaluation indicators are clearly defined, and the appendix supplements the bootstrap statistical significance analysis to ensure the reliability of the results.

**Weaknesses:**

1. The paper finds significant differences in the position and width of the "fusion band" of different models (e.g. early wide fusion of Qwen2-VL, late decentralized fusion of LLaVA-1.5), but does not analyze in depth the direct relationship between architectural design and fusion mode. For example, how do architectural differences such as "visual adapter" of Qwen2-VL, "CLIP + Vicuna splicing architecture" of LLaVA-1.5, and "MoE decoder" of DeepSeek-VL2 specifically affect the timing and scope of fusion? Existing discussions only mention "architectural differences", and lack quantitative or qualitative relevance arguments.
2.  MINT is based on a single level of hidden state patching, but does not explain how to handle the impact of cross-level fusion interactions - if a layer of fusion relies on the modal representation of the preceding layer, will a single layer of patching underestimate or misjudge the fusion key layer?
3. Text grounding experiments show that text embeddings are weak and limited to early layers of visual information, and the model relies mainly on direct visual attention. But the paper does not further explore "whether enhancing text grounding can improve the fusion effect" - for example, by fine-tuning the text encoder to carry visual information in the first order, will it reduce the "fusion band" range or improve the fusion accuracy?

**Questions:**

This paper has made valuable contributions to the field of VLM multimodal fusion interpretability. The MINT method and the discovery of "fusion band" provide new tools and perspectives for understanding the fusion mechanism. If experiments and analysis can be supplemented to address the above shortcomings, and the generalization, depth and practical value of the conclusions can be further improved, this work will be more in line with the publication standards.

---

> ### Author Response · Authors · 2025-12-01
>
> We sincerely thank the reviewer for their constructive feedback. We are encouraged by the reviewer's recognition of MINT's "theoretical significance" and rigorous evaluation. To address the reviewer's concerns regarding practical utility and architectural drivers, we conducted a new targeted intervention experiment.
>
> **About Practical Utility: MINT-Guided Surgical Repair**
>
> The reviewer asked whether enhancing specific layers could improve fusion. To validate this, we used MINT to guide a **Precisely Localized LoRA Fine-Tuning** on Qwen2-VL to correct its hallucination failures (predicting 'Yes' for absent objects).
>
> We compared fine-tuning the **MINT-identified "Fusion Band" (Layers 14–16)** against a **Baseline (Shallow Layers 1–7)**. We evaluated correction rates on known failures and generalizability on the full benchmark (5,000 samples).
>
> | LoRA Strategy | Layers | Params | Correction Rate | General Acc. | Gain |
> | :--- | :---: | :---: | :---: | :---: | :---: |
> | Original Model | N/A | N/A | 0.00% | 94.04% | - |
> | Baseline (Shallow) | 1–7 | $2.33\times$ | 77.63% | 94.36% | +0.32% |
> | **MINT-Guided** | **14–16** | **$1\times$** | **100.00%** | **99.98%** | **+5.94%** |
>
> **Result:** MINT-guided repair achieved a **perfect correction rate (100%)** and solved the general task (~99.9% accuracy) using **less than half the parameters** of the baseline. This empirically proves that MINT correctly identifies the most efficient architectural bottleneck, transforming interpretability into a tool for surgical model repair.
>
> **About Single-Layer Patching vs. Cumulative Effects**
>
> We agree that information fusion is a cumulative process. However, our surgical fine-tuning experiment (Section 5.5) provides strong empirical validation for our single-layer diagnostic approach.
>
> This experiment revealed:
>
> * Fine-tuning a broad, *shallow* range (Layers 1-7)—representing the cumulative buildup—was less effective (77.63% correction).
> * In contrast, a surgical intervention on *only* the 3-layer "Fusion Band" identified by MINT (Layers 14-16) achieved a **perfect 100.00% correction** using less than half the parameters.
>
> This demonstrates that while information builds up cumulatively, MINT successfully locates the **"Causally Decisive Point"**, which is the specific layers where the representation becomes *sufficient* to determine the output and where intervention is most efficient.
>
>
> **About Architectural Drivers of Fusion Bands**
>
> We agree this is a crucial point. We argue that MINT provides the first diagnostic framework to formulate testable hypotheses about *why* these fusion patterns differ across architectures. Based on our findings, we will add these hypotheses to our Discussion section:
>
> * **Connector Complexity vs. Timing:** The fusion pattern appears linked to the connector. We hypothesize simple connectors (like LLaVA's MLP) pass raw features, forcing the decoder to perform alignment and resulting in the "Late Fusion" pattern we observed. Conversely, sophisticated adapters (like Qwen's) deliver highly pre-processed features, enabling "Early Fusion".
>
> * **Localization vs. Robustness:** Our diagnostics revealed a potential tradeoff. DeepSeek-VL2's highly localized and early fusion (peaking at Layer 4) correlates with its **distinct performance characteristics** on geometric reasoning. Our results show patching in correct visual information is **insufficient to alter its output** (a low Flip Rate across all layers). This suggests this efficient, concentrated fusion may lack the redundancy needed for such complex tasks.

---

### Official Review · Reviewer_os6o · 2025-11-01

**Soundness:** 1
**Presentation:** 2
**Contribution:** 2
**Rating:** 2
**Confidence:** 4

**Summary:**

Vision-Language Models (VLMs) integrate visual and textual information through complex internal mechanisms that remain poorly understood. This paper presents MINT (Multimodal INtervention Tracing), a causal framework for analyzing how and where information fusion occurs within multimodal decoders. Instead of relying on correlation-based probing, MINT performs representation patching, systematically swapping hidden states between different runs, to trace the causal influence of visual and textual inputs layer by layer. Through experiments on multiple VLMs, the authors identify characteristic fusion bands and model-specific fusion patterns, showing how visual information overrides language priors. The additional application shows that the framework also allows the diagnosis of failure cases.

**Strengths:**

1. The introduction of MINT moves beyond correlation-based probing toward causal tracing through hidden-state patching. The idea of swapping intermediate representations is simple and conceptually transparent. It makes sense that changes in output can be causally attributed to specific layers or modalities.
2. The study evaluates multiple major VLMs across benchmarks, helping with meaningful cross-model comparison. The identification of the proposed "fusion bands", and the detailed mapping of which layers contribute to visual grounding or fail, provide generalizable findings and valuable insights for targeted model improvement.

**Weaknesses:**

1. While the framework is clearly presented, its core technique, hidden-state patching, is not novel. The contribution primarily adapts this existing approach to the multimodal setting rather than introducing a fundamentally new causal mechanism. The study could be made more compelling by extending the intervention beyond image features to also include text patching, enabling a fuller analysis of bidirectional information flow between modalities.
2. The paper claims to present the first empirical map of the "fusion band", yet the term itself is newly introduced and lacks grounding in prior literature. Moreover, the evaluation framework relies entirely on in-house binary tasks and custom metrics (override accuracy, flip rate, and failure depth), which are defined within this paper. This makes it difficult to compare results or validate the claimed novelty against established evaluation standards.
3. All main experiments adopt a classification-style prompt, focusing on binary outputs rather than analyzing finer-grained probability shifts that could reveal more nuanced causal effects. This, in fact, again places this work among output-level analyses rather than deeper investigations into the internal latent representations of VLMs. Consequently, the experimental setup feels shallow and does not fully leverage existing multimodal benchmarks. The experiments are also incomplete and inconsistent. For instance, LLaVA is not included in all analyses. Also, there is no direct comparison with existing interpretability or causal probing methods, despite their discussion in the related work section.

**Questions:**

What are the new benefits MINT offers compared with earlier methods? The main concern is its lack of connection and comparison with past studies. The novelty, depth, and coverage are also limited. While minor, use abbreviations correctly. For example, the "Vision-Language Model (VLM)" repeats multiple times.

---

> ### Author Response · Authors · 2025-12-01
>
> We thank the reviewer for recognizing the "conceptual transparency" and "generalizable findings" of our work. We appreciate the reviewer's critical feedback and offer the following clarifications.
>
> **About Methodological Novelty**
>
> We respectfully clarify that adapting patching from unimodal models (single-token interventions) to VLMs (distributed visual sequences) is a **non-trivial methodological leap**. MINT is the first framework to perform stable, **sequence-level interventions** (i.e., patching hundreds of tokens), a prerequisite for VLM decoder tracing that addresses a key feasibility question of decoder robustness. This adaptation was essential to construct the **first causal "fusion map"** of the decoder and discover phenomena like "Late Activation," which standard methods cannot access.
>
> **About the Scope of Intervention (Text Patching)**
>
> We respectfully clarify a misunderstanding. The suggestion to "include text patching" is **already a core part of our work**. **Experiment 2 (Section 5.2: "When Do Text Embeddings Become Visually Grounded?")** is entirely dedicated to this, with results visualized in **Figures 5 and 9**. This confirms MINT is a **fully bidirectional framework** for tracing both visual and textual information flow. We will revise the text to make this capability more prominent.
>
>
> **About Terminology and Causal Metrics**
>
> We argue these are necessary to describe novel mechanistic phenomena that standard performance metrics cannot capture.
>
> * **"Fusion Band" Terminology:** We introduced "Fusion Band" as necessary vocabulary for the specific layer-wise fusion phenomenon we discovered. This follows standard scientific practice in interpretability (e.g., "Knowledge Neurons," "Induction Heads") of defining terms for newly identified internal mechanisms.
> * **Causal Metrics:** Our metrics ("Flip Rate," "Override Accuracy") are not arbitrary. They are direct adaptations of the **Average Treatment Effect (ATE)**, a standard measurement in causal inference. These metrics are essential to quantify the *causal effect* of internal components, whereas standard benchmarks (like Accuracy) only measure *performance outcomes*.
>
> **About Experimental Depth and Consistency**
>
> We would like to clarify these two points:
>
> 1.  **On Experimental Depth:** Regarding the "binary" task, our analysis is not based on the final output but tracks the underlying **continuous logit probability shifts**. The constrained Yes/No format is an intentional probe, standard in mechanistic interpretability (e.g., Patchscopes), to strictly **isolate the causal effect** of the patch from the noise of open-ended decoding.
>
> 2.  **On Model Consistency:** The reviewer is correct that LLaVA was omitted from Experiment 2 (Text Grounding). This was due to a technical compatibility issue with our caching mechanism for that specific architecture. We have since resolved this and **will include the full LLaVA results** for this experiment in the final manuscript to ensure complete consistency.
>
> **About Comparison with Baselines (New Benefits)**
>
> MINT offers a critical methodological advantage over the two dominant paradigms mentioned in our Related Work: **Linear Probing** and **Attention Analysis**.
>
> * **MINT vs. Linear Probing (Causality):** Standard probing classifiers only detect if information *exists* in a layer (correlation). MINT detects if that information actually *drives the output* (causation). By intervening directly, MINT filters out "inert" information that correlates with the input but is ignored by the decoder, a distinction probing cannot make.
> * **MINT vs. Attention (Depth):** While attention maps visualize *spatial* focus ("where" the model looks), they offer limited insight into the *depth* of integration ("at which layer" the decision is finalized). MINT provides a unique "z-axis" map of the fusion process, verifying the functional role of hidden states rather than relying on attention weights, which often dissociate from model predictions.
>
> *Minor:* We have corrected the redundant abbreviations of "VLM" throughout the manuscript as requested.

---

### Official Review · Reviewer_d5Wo · 2025-11-11

**Soundness:** 2
**Presentation:** 2
**Contribution:** 1
**Rating:** 2
**Confidence:** 4

**Summary:**

This paper introduces MINT, a novel framework designed to construct a causal map of multimodal processing in vision-language models (VLMs) by leveraging hidden-state patching techniques. The authors conduct experiments to investigate the internal mechanisms of LLaVA-1.5-7B, DeepSeek-VL2-Tiny, and Qwen2-VL-7B. And the results reveal some insights on the 'fusion band' of VLMs.

**Strengths:**

- Clear and Well-Structured: The paper is well-organized, with detailed explanations of the preliminary, intuition, and methodology.

- Extensive Evaluations: Three representative VLMs are investigated with the proposed framework, accompanied by comprehensive analysis and discussion.

- Interesting Findings: The experiment results offer some findings on the multimodal processing of VLMs.

**Weaknesses:**

- My major concern is that the core techniques incorporated in MINT are based on the patching method introduced in the previous work, **Patchscopes**. While the derivation is clear and well-presented, it does not introduce fundamentally new concepts but rather applies existing methods in a different context.

- The models used in the experiments are somewhat outdated — all evaluated models were released over 12 months; a more comprehensive evaluation using such recent and stronger VLMs would strengthen the manuscript.

- While they find variation across models, it’s unclear why some models adopt early vs late fusion. Are these choices by design (architecture) or emergent? Are they correlated with performance tradeoffs?

**Questions:**

See Weaknesses

---

> ### Author Response · Authors · 2025-12-01
>
> We thank the reviewer for acknowledging our paper's clear structure, extensive evaluations, and interesting findings. We address the reviewer's three main concerns below.
>
> **About the Methodological Novelty of MINT**
>
> We respectfully clarify that adapting patching from unimodal models to VLMs represents a **non-trivial methodological leap**. This is MINT's core contribution.
>
> 1.  **Different Intervention Scope:** Methods like Patchscopes intervene on single or sparse tokens in LLMs. In contrast, VLM visual information is **highly distributed across hundreds of tokens** (e.g., 576 tokens). A simple single-token patch, as used in LLMs, is insufficient and yields no observable causal effect in this context. MINT is the first framework designed to perform stable, **sequence-level interventions** (i.e., replacing the *full* visual patch sequence).
> 2.  **Addressing a Key Feasibility Question:** A critical unknown before our work was whether the decoder would even remain **robust** or simply fail (e.g., output incoherently) when subjected to such a **massive, sequence-level state substitution**. Demonstrating that this is not only feasible but that the decoder is remarkably robust is a key technical finding in itself, and a prerequisite for any causal tracing in the VLM decoder.
> 3.  **Enabling New Discoveries:** This sequence-level intervention framework is precisely what enabled our primary scientific contribution: constructing the **first causal "fusion map"** of the decoder. It allowed us to trace visual information flow *within* the LLM backbone (not just the encoder/projector) and thus discover phenomena like "Late Activation," which are inaccessible to standard probing methods.
>
> In summary, the novelty lies in designing a sequence-level intervention robust enough for the VLM decoder, which in turn unlocked the ability to causally analyze its internal fusion mechanisms.
>
> **About the Selection of Models and Recency**
>
> We appreciate the concern regarding model recency. We want to clarify that our initial model selection (LLaVA-1.5, DeepSeek-VL2, Qwen2-VL) was **deliberately chosen to represent key architectural archetypes** rather than just release dates, as our focus is on mechanisms. These models exemplify dominant design paradigms (e.g., simple MLP connectors, complex vision adapters) that remain prevalent.
>
> To **further validate MINT's utility and directly address your concern**, we have extended our evaluation to include the state-of-the-art **Qwen2.5-VL-7B**.
>
> Applying MINT to this newer architecture revealed a novel phenomenon distinct from its predecessor (Qwen2-VL). Instead of a localized "fusion band," Qwen2.5-VL exhibits a pattern of **"Sustained Visual Integration,"** where visual interventions remain causally effective throughout nearly the entire decoder depth.
>
> This new finding is significant: it suggests SOTA architectures may be evolving towards maintaining higher visual plasticity, and it **demonstrates MINT's capability as a sensitive diagnostic tool** that can capture these critical architectural evolutions. We have incorporated this new analysis into the updated manuscript (e.g., Figure 3 and Section 5.1).
>
> **About the Implications of Fusion Variation**
>
> We agree this is an incisive question. We argue that MINT provides the first diagnostic framework necessary to formulate testable hypotheses about *why* these variations exist. Our findings allow us to posit two key hypotheses:
>
> 1.  **Link to Architecture:** The "fusion band" location appears determined by the vision-language connector. We hypothesize that simple connectors (like LLaVA's MLP) pass raw features, forcing the decoder to perform alignment and resulting in the "Late Fusion" pattern we observed. Conversely, sophisticated adapters (like Qwen2-VL's) deliver highly pre-processed features, enabling the "Early Fusion" signature.
>
> 2.  **Link to Performance Tradeoffs:** Our diagnostics revealed a potential tradeoff. DeepSeek-VL2's highly localized and early fusion (peaking at Layer 4), while efficient, correlates with its **distinct performance characteristics** on complex spatial reasoning tasks. This model shows a low Flip Rate across all layers, indicating that patching in correct visual information is insufficient to resolve its errors, a phenomenon likely tied to its architectural design. In contrast, Qwen2-VL's broader fusion window appears crucial for the "late activation" of its reasoning abilities (around layer 15), linking this mechanism to robustness on such tasks.
>
> While a full investigation of these links is key future work, our framework provides the essential map to guide it. We will explicitly incorporate these hypotheses into the revised Discussion (Section 6).

---

### Note · Authors · 2026-01-06

I have read and agree with the venue's withdrawal policy on behalf of myself and my co-authors.